# Preliminary Experimental Results of Context-Aware Teams of Multiple Autonomous Agents Operating under Constrained Communications

Jose Martinez-Lorenzo [1,*], Jeff Hudack [2], Yutao Jing [1], Michael Shaham [1], Zixuan Liang [1], Abdullah Al Bashit [1], Yushu Wu [1], Weite Zhang [1], Matthew Skopin [1], Juan Heredia-Juesas [1], Yuntao Ma [1], Tristan Sweeney [1], Nicolas Ares [1] and Ari Fox [1]

[1] College of Engineering, Northeastern University, Boston, MA 02115, USA
[2] US Air Force Research Laboratory, Rome, NY 13441, USA
[*] Correspondence: j.martinez-lorenzo@northeastern.edu

**Abstract:** This work presents and experimentally tests the framework used by our context-aware, distributed team of small Unmanned Aerial Systems (SUAS) capable of operating in real time, in an autonomous fashion, and under constrained communications. Our framework relies on a three-layered approach: (1) an operational layer, where fast temporal and narrow spatial decisions are made; (2) a tactical layer, where temporal and spatial decisions are made for a team of agents; and (3) a strategical layer, where slow temporal and wide spatial decisions are made for the team of agents. These three layers are coordinated by an ad hoc, software-defined communications network, which ensures sparse but timely delivery of messages amongst groups and teams of agents at each layer, even under constrained communications. Experimental results are presented for a team of 10 small unmanned aerial systems tasked with searching for and monitoring a person in an open area. At the operational layer, our use case presents an agent autonomously performing searching, detection, localization, classification, identification, tracking, and following of the person, while avoiding malicious collisions. At the tactical layer, our experimental use case presents the cooperative interaction of a group of multiple agents that enables the monitoring of the targeted person over wider spatial and temporal regions. At the strategic layer, our use case involves the detection of complex behaviors, i.e., the person being followed enters a car and runs away, or the person being followed exits the car and runs away, which require strategic responses to successfully accomplish the mission.

**Keywords:** SUAV; multi-layer architecture; context aware; decentralization; complex behavior

## 1. Introduction

Recent advancements in the fields of Artificial Intelligence [1–7], Machine Learning [8–11], Robotics [12–16], and Signal Processing [17–19] have provided humankind with unique set of tools that, for the first time in history, have the potential to address some of the most important problems existing in the field of group autonomy of unmanned systems [20]. These advancements naturally create a need for real-time systems and algorithms capable of providing decentralized multi-sensor classification and multi-agent coordination in missions subject to constrained communications, [21,22]. This is especially important for teams of agents performing non-centralized surveillance with non-scheduled coordination, [23,24]. An important challenge of such missions is that current positioning systems are neither fast nor effective when dealing with high-dimensional data. Nowadays, group autonomous systems require either direct human control of many systems [25,26], contract and auction techniques [27,28], or coalition methods [29–31]. The latter are heavily dependent on the communications channel, which is often constrained in many realistic scenarios. This connects to an additional challenge, since conventional systems do not fully take into

consideration the instantaneous performance of the communications network or sensing platforms in order to dynamically adapt task priorities in multi-agent missions. Other approaches based on Markov Decision Processes do not scale linearly with the number of agents and states, and they often result in a slow reaction to unexpected events, [32–36]. Finally, another important factor is that SUAS hardware platforms are severely constrained in terms of on-board processing capability, as well as communication range, and state-of-the-art networks do not ensure that the data are efficiently shared among other members of the SUAS swarm, or relayed in a timely manner over multiple hops to the ground station (if used) during network operation [37,38].

This paper describes and experimentally validates the framework—hardware, software, and system of systems architecture—used by our context-aware, distributed team of Small Unmanned Aerial Systems (SUAS), which can operate in real-time, in an autonomous fashion, and under constrained communications. Our framework relies on a three-layered approach: (1) An operational layer (fast temporal and narrow spatial scale; partially mimicking functionality of human's peripheral nervous system)—here, a single agent performs on-board detection, localization, classification, identification, tracking, and following while avoiding malicious collisions; this layer relies on hardware and software that enable the fusion and sparsification in real-time of 4D full motion video, 4D millimeter wave radars, 4D infrared cameras using Deep Learning and 4D (space + time) compressive sensing (CS). (2) A tactical layer (intermediate temporal and spatial scale; partially mimicking functionality of human's muscular system)—here, a group of multiple autonomous agents collaborate to jointly perform a complex task that cannot be executed by a single agent due to their spatial (navigation) and temporal (perception) limitations. (3) A strategical layer (slow temporal and wide spatial scale, partially mimicking the functionality of the endocrine system)—here, teams of multiple autonomous agents cooperate to jointly perform a multi-step complex task that cannot be executed by a group of autonomous agents due to their spatial (navigation), temporal (perception), and energy (endurance) limitations. These three layers are coordinated by an ad hoc, software-defined communications network, which ensures sparse, but timely delivery of messages amongst groups and teams of agent even under constrained communications.

This architecture facilitates the optimization of each layer independently, providing a solution to the common scalability problem; namely, the number of platforms does not scale in a linear fashion, but in an exponential way. Therefore, diverse missions involving complex decision making in unstructured environments can be executed with this unique framework. Furthermore, the proposed architecture allows current leader–follower formations to be overcome, which often use static communications, and scripted rule-based behaviors [39,40]. Our proposed approach demonstrates fluid distributed maneuvers, collaborative schemes, and flexible coordination.

## 2. System Architecture

The hierarchical architecture adopted by our autonomous multi-agent system is the one presented in Figure 1. From an operator perspective, as shown at the top of the figure, a general mission is specified for a team of agents that must organize themselves to accomplish a particular mission. In this paper use-case, the general mission is defined as *search and monitor people in a given region*. Based on this mission, an offline planner parses a multi-layer policy (controller) to each agent in the network using a top-to-bottom approach. The latter leverages the use of a set of memory banks, which resemble the different types of memories used by the human body, including: (i) long-term strategic memory, which covers spatial priming memory and temporal procedural memory; (ii) long-term tactical memory, which covers spatial semantic memory and temporal episodic memory; and (iii) short term memory and sensory memory.

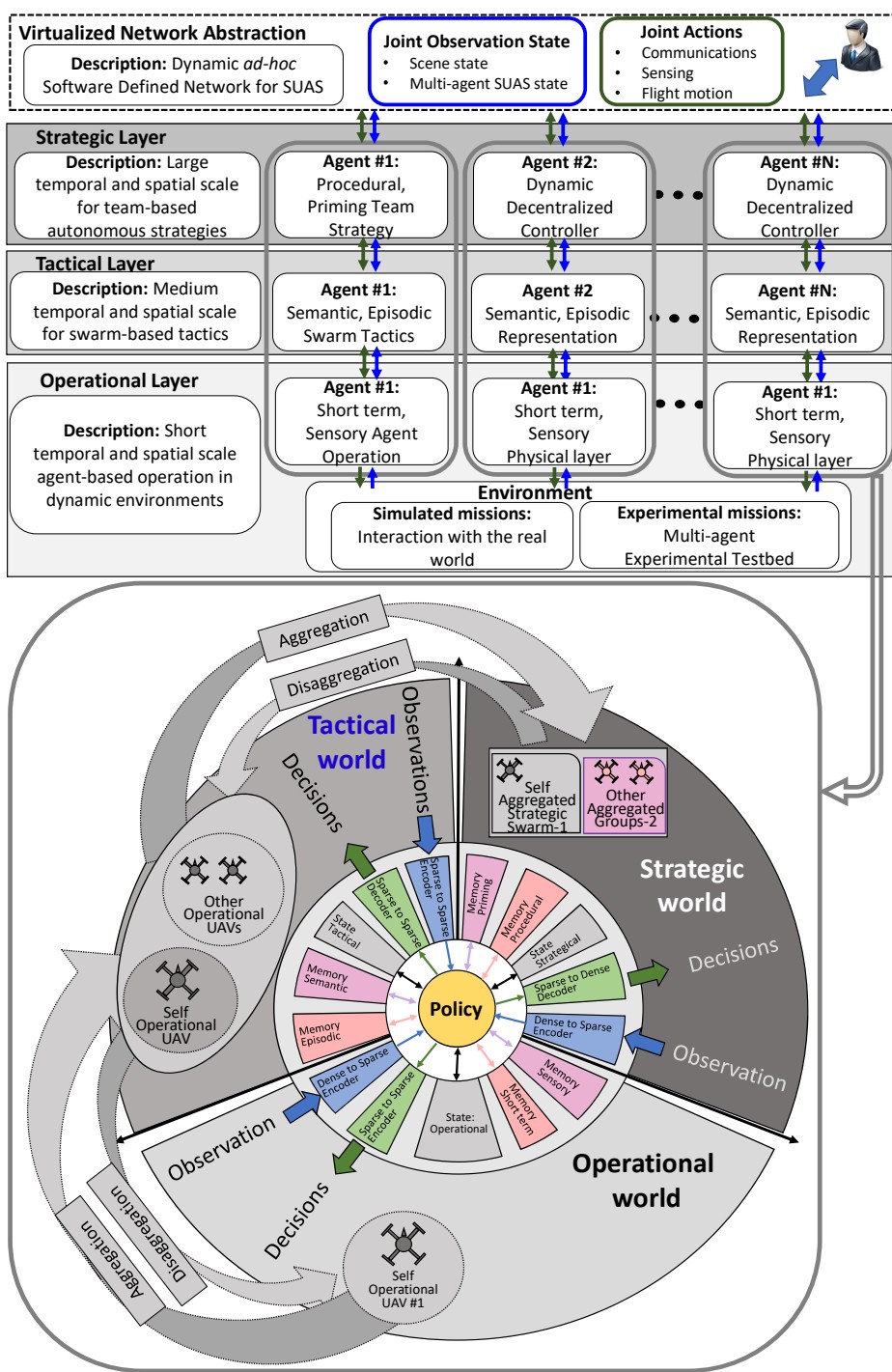

**Figure 1.** Decentralized autonomous systems: (**top**) team operator perspective; (**bottom**) agent in the team perspective.

The strategic memory is used to load the initial strategic policy, as well as the type of decisions and observations available for the team of agents at this level. Similar functionality is provided to the tactical and operational memory, regarding decisions, observations, and policies at its corresponding layer. In particular, the tactical layer is responsible for the algorithms for group control, which is built upon the operational controllers, and manages the exchange of messages among the agents within the team, including the messages used to start the tactical maneuver, control messages to fuse the information from multiple agents during the tactical execution, the messages used to terminate the tactical maneuver, and the messages used to perform the hand-off from a tactical team to another tactical team.



From an agent perspective, our architecture enables each unmanned system to reason about its own operation, its tactical relationships with a subgroup of agents with whom it is cooperating in a joint task, and its strategical contribution to the overall mission. This perspective is shown at the bottom part of Figure 1, and a thorough description is described in the next section.

### 3. Multi-Layer Perception

As shown in Figure 2, the agents (SUAVs) in our network can be equipped with three different type of perception sensors: a 4D RGB camera, a 4D Infrared camera, and a 4D mmWave radar. At the operational level, the raw data of the three sensors is parsed into a Vector Processing Unit, which runs a fine-tuned Convolutional Neural Network to perform the sensor fusion and to provide a sparse representation of the scene. As can be seen on the top area of Figure 2, our dense-to-sparse perception module is capable of outputting sparse information about the scene at a 5 Hz rate—an enhanced frame rate of 100 to 1000 Hz should be achieved with our current architecture. This output contains a list of targets in the scene (e.g., person, car, etc.), a classification confidence level for each target, targets' bounding boxes in 2D, targets' ranges from the agent, targets' angular location relative to the agent's orientation, as well as 4D GPS geolocation of both targets and agent. At the tactical level, medium-priority observations involving other agents within the same group, jointly performing a particular activity, are sparsely parsed through the ad hoc network in an asynchronous fashion at a reduced average rate (~0.01 Hz per mission). Similarly, at the strategical level, top-priority observations of events that require an update on the team strategy are parsed through the ad hoc network in an asynchronous fashion at a very low average rate (~0.005 Hz per mission).

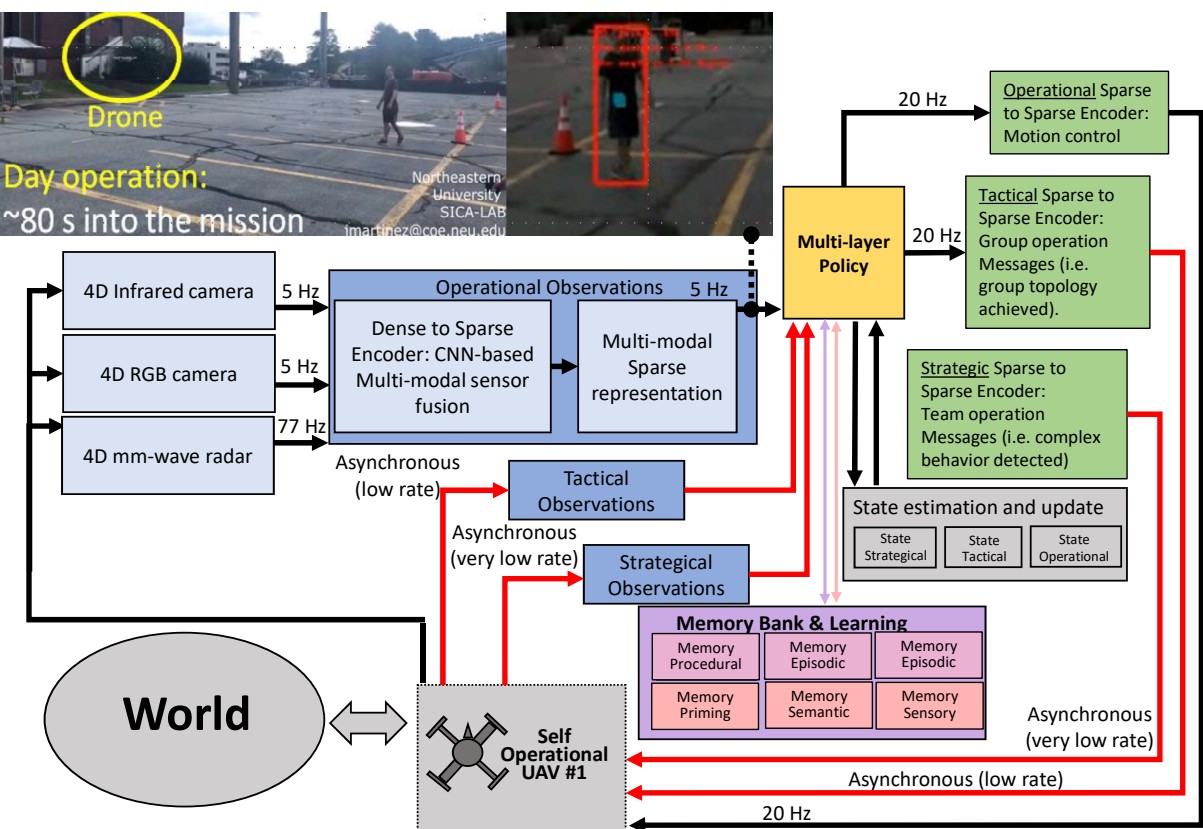

**Figure 2.** Architecture of multiple sensors fusion with Convolutional Neural Network. Inference results of the fusion module feed into our drone policy, which controls the motion of the drone or performs complex behaviors and outputs environmental prediction.

## 4. Multi-Layer Policy

The vector targets at the operational layer are synchronously parsed to our multi-layer policy block (see Figure 2), which uses a sparse-to-sparse motion controller to generate motion trajectories based on the agent's particular state. At the operational level, each drone can simultaneously be active in one or more of the following operational states: *idle/sleeping*, *takeoff/landing*, *searching*, *following*, *tracking*, *navigating to a GPS location*, *returning to base*. The latter sparsification at the operational state affords scalability of the multi-layer policy. At the tactical level, the multi-layer policy handles the information received either by its own operational observations or from another member of its tactical group. The policy enables asynchronous coordination of the group of agents at the tactical level. When a group of agents is not able to continue a particular group activity, the strategic policy may be able to recruit another group of agents that can finalize the mission in a suitable fashion. The strategic policy observes and controls the strategic perception and actuation channels. The use-case described below will clearly emphasize the type of observations and actions that are provided for each one of the components of the multi-layer policy.

## 5. Multi-Layer Decisions

The multi-layer policy creates a sparse vector that encodes the actions needed at the strategical, tactical, and operational level. In the latter, the *operational sparse to sparse encoder* shown in Figure 2 generates the signals needed to control the lower-level motion controller. Our system follows a control approach similar to the one presented in [41]. Specifically, once the target is recognized and localized, the drone will change its state to follow or track the object and update its position, based on the change in position obtained from its own observations. Position updates can be made by sending the flight controller either local velocity setpoints or local or global position setpoints. By receiving the angle and the distance of the target relative to its current position and orientation, the controller will decide how much to rotate and how to adjust its position. Various uncertainties, such as external forces (e.g., wind), can affect the motion of the drone, and the flight controller needs to be able to compensate. Changes to the controller can also come from communication with other drones or other swarms, or from recognizing complex behavior or patterns. When a drone recognizes certain behavior occurring among the objects it is seeing (e.g., a person entering a car), it can communicate to the other drones to change their state (e.g., to return home) and to other swarms to begin or change their mission.

## 6. Results

At the top of the system is the mission controller, which controls the subsystems in an attempt to achieve the swarm's objective. Mission objectives for the swarm of drones are typically defined by an area of exploration and a searching objective, e.g., finding survivors in a disaster-struck area. To maximize the ability to search an area and understand the environment, the swarm needs to be divided into a specific number of subswarms depending on the environment and objective. For example, the area of exploration can be partitioned into different sections, each of which is searched by a different subswarm. If needed, subswarms can decide to split into smaller subswarms depending on what is best for the environment they are in. For example, a subswarm may encounter a building or multiple buildings, and may need to split up to search these newly encountered parts of the environment. At the smallest scale in this system, individual drones make observations and act on them based on a learned policy. Communication with other drones in the subswarm occurs depending on its observations. Finally, a mission ends when the mission controller sends the signal that the mission is over, either based on time or observations.

This paper brings up the proof of concept of experimenting with drones in navigating, tracking, following, and landing modes with a swarm of ten drones, as represented in Figure 3. In this experiment, swarm-1, swarm-2, and swarm-3 have three, four, and three drones, respectively, which are participating in the mission, as is represented in detail in Figure 4. The tasks for swarm-1 are detecting, following the person until he/she enters into

the car, which has been defined as a complex behavior. At this point, the *return-to-launch* command is sent to instruct all members of the swarm to return to base, and the *take-off* command is sent to the drones in swarm-2. The tasks for the second swarm are flying to the predefined GPS locations, and starting tracking, which means facing to direction where the car moves. The drones keep on tracking until the car stops and the person gets out of the car, which also has been defined as a complex behavior, or the *land* command is received from their peers or the ground control station. At this stage, if any of the drone detects the complex behavior, i.e., a person and a car is present, it immediately sends the *take-off* command to swarm-3 to fly towards the detected person. The tasks for swarm-3 are flying to the GPS position sent from the drone, and to start following the person. Swarm-3 identifies the person and maintains a constant distance from them before landing. The detailed sequence of the mission, along with some frames captured by the cameras of the drones incorporating their perception, is shown in Figure 5.

The perception in swarms 1 and 2, which leads to the detection and tracking of the person and the car, is performed by a computer vision algorithm based on the pretrained MobileNet-SSD convolutional neural network [42], which is adapted to detect only the targets of interest. The front RGB camera captures the target within a rectangle and, for this experiment, focuses only on the first target that appears on the scene. When the midpoint of that rectangle appears deviated from the center of the camera, the flying algorithm causes the drone to rotate until the target center point meets within the threshold of the tracking pattern. Meanwhile, the depth camera measures the average distance to the target. When the distance increases over a given value $D_0$, the flying algorithm pushes the drone closer to the target and vice-versa.

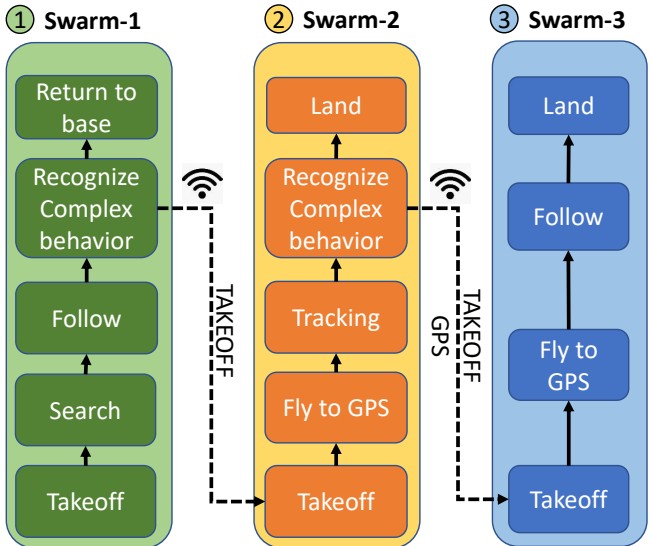

**Figure 3.** Multi-Swarm mission description.

On the other hand, the perception in swarm-3, which leads to following at a constant distance from the person, is performed by extracting the range distance from the radar 3D point cloud, followed by negative feedback to the UAV fight controller and moving $-(R - R_0)$ meters in the range direction. This is the direction of the front view of the camera, where $R_0$ is the constant following distance and $R$ is the range distance detected by the radar.

In order to ensure that all perception components (detection, localization, classification, and tracking) run in real time, they are executed at 10 frames per second, which is necessary to send off-board commands to the drone for real-time adaptation to the unstructured environment. The quad-core Intel Atom processor, together with a neural computer stick

installed in each drone, enables fast data analysis and signal processing to achieve the above-mentioned requirements.

All individual pieces of the system, such as the autonomous flight to a GPS location, the exchange of messages and commands, the optical and radar detections, the sensor fusion-based tracking, etc., were tested, tuned, and validated several times, with simulators and in the real scenario, and were appended little by little until building up the whole autonomous system.

Figure 6 shows the performance of the current experiment. For each swarm, the right column with light colors indicates the minimum number of drones expected to perform the specific behavior, and the left column displays the actual number of drones that performed each of the indicated tasks. For swarm-1, all drones took off and started their search paths. Two out of the three drones found and detected the person, made the decision to follow them, and finished the task successfully; they neither lost the target while following them, nor hit any obstacle accidentally. The other drone did not find any person, which is acceptable, since it was carrying out the search over a region where there were no people at that time. Once one of the drones recognized the complex behavior of the person entering the car, it sent the *return-to-launch* command to all drones in its swarm, and the *take-off* command to swarm-2, which activated the beginning of the mission corresponding to this swarm. All drones in swarm-1 returned to base successfully. In swarm-2, three out of four drones took off and reached their predefined GPS points, rotated around their own vertical axis and statically tracked the car as expected. One of these three drones recognized the complex task of detecting the targeted person entering into the car. At this point, this drone broadcast the *land* command to the members of its own swarm, leading all drones to successfully land, and the *take-off* command to swarm-3. Finally, in the swarm-3, one out of three drones took off and performed the tasks of flying to the GPS point, track, and follow successfully. After a time-out, it landed; however, the other two did not take off as expected, probably due to communications issues or package loss, which prevented them receiving the *take-off* command.

It is important to notice that not all the drones in each swarm are expected to perform every single task within their corresponding sequence. The failure of a drone performing a task can be covered by its peers. The overall mission is successful if the whole sequence is executed, which in this example was to find and follow a person, identify when they enters a car, track the car, identify the person exiting the car, and follow that person again. The proposed mission in this paper was, therefore, successfully executed.

*Discussion*

During testing, it was observed that for tracking in negative areas—such as the dark side of the car—communication antenna orientation, wind speed, sensor calibration, and distance between drones result in packet loss, affecting the detection, navigation, and tracking performance for the swarms performing a desired task.

In addition, when multiple targets are captured by the camera of the drone, such as several people or cars in the same frame, some constraints may limit the drone operation, leading to possible false tracking. In the presented case, the person or car that first appears in the drone's field of view is considered the main target, and it must track them without losing or switching targets. However, if two people appear on the scene too closely or overlap each other, it is possible that the tracker may switch their main target, leading to a failed mission. In future experiments, where the requested mission will be much more complex than the current experiment, it will be crucial for the team of SUAS to obtain as much as information as possible from the outside environment. For these cases, a multiple objects tracking (MOT) approach is expected to be more reliable in realistic scenarios. This functionality, which will be vital for a team of SUAS perceiving a large-scale environment, is already available with current online MOT methods, such as deepSORT, MHT_bLSTM, and OneShotDA, benefiting the extensibility of our approach for more complex missions [43,44].

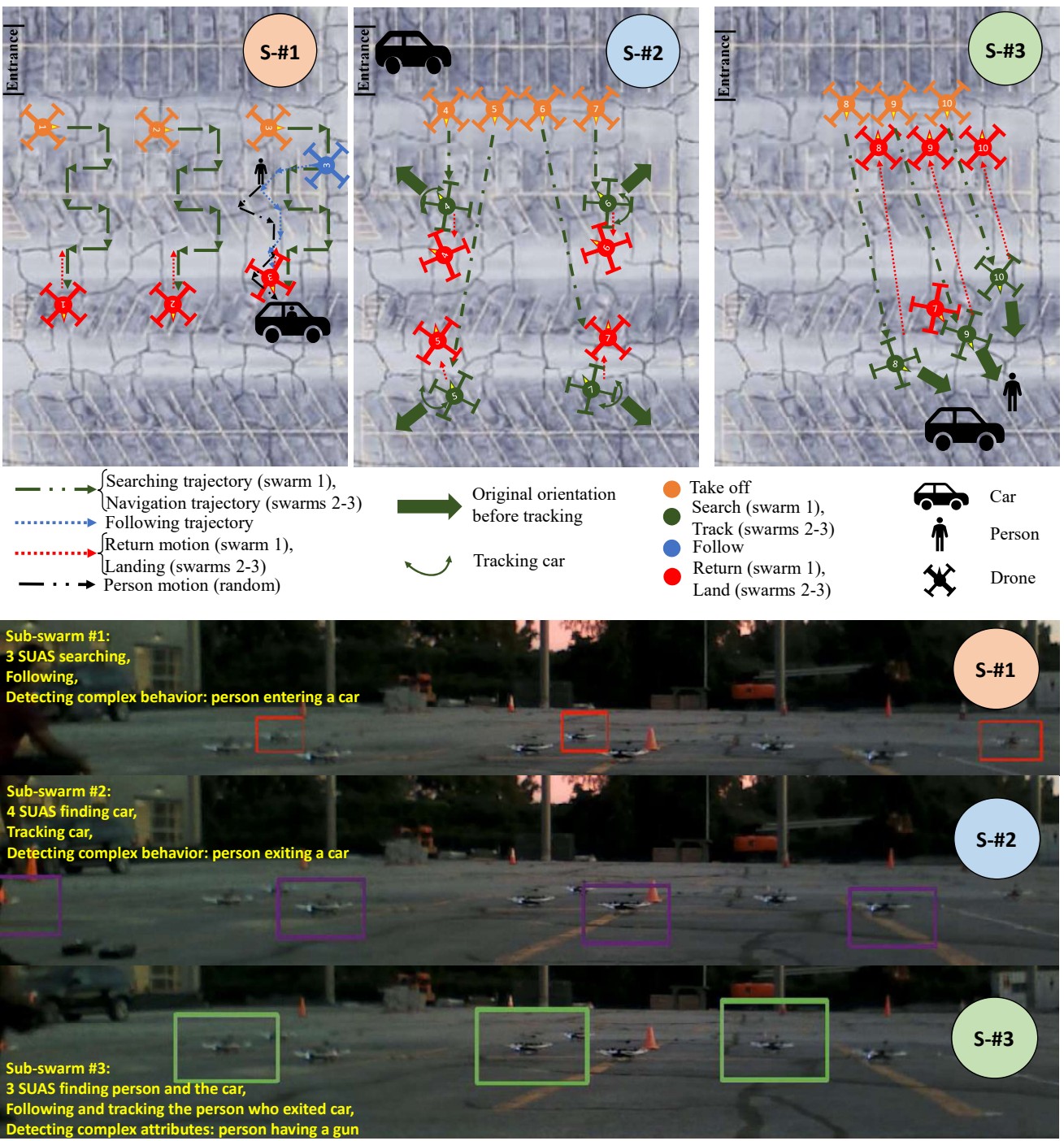

**Figure 4.** Autonomous Multi-swarm demonstration. (1) Sub-swarm #1 searches for a person, who is followed until it enters a car; (2) sub-swarm #2 tracks the car around the "simulated road"; (3) sub-swarm #3 tracks the person after they exit the car.

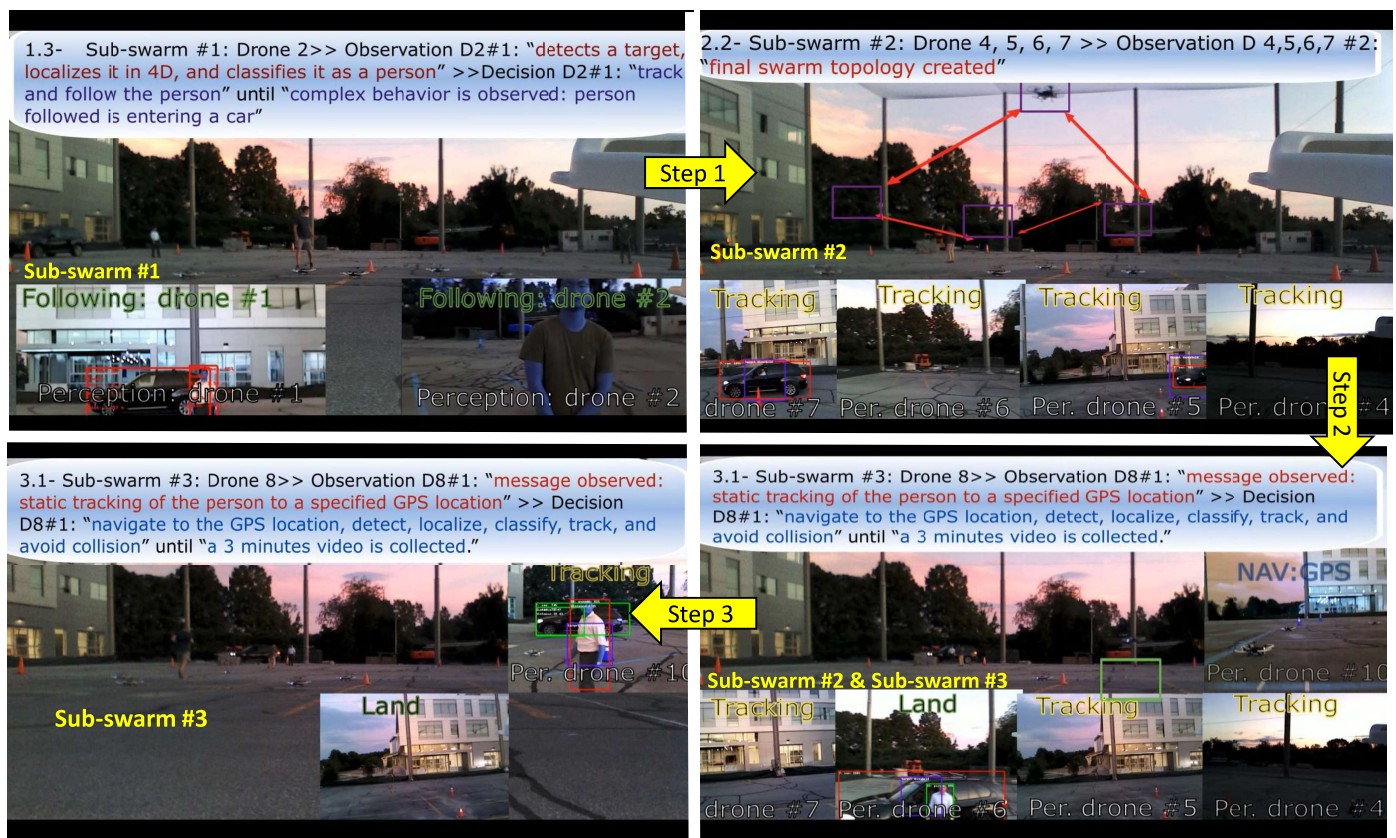

**Figure 5.** Overall sequence of the whole mission recorded by a stationary ground camera and a drone camera.

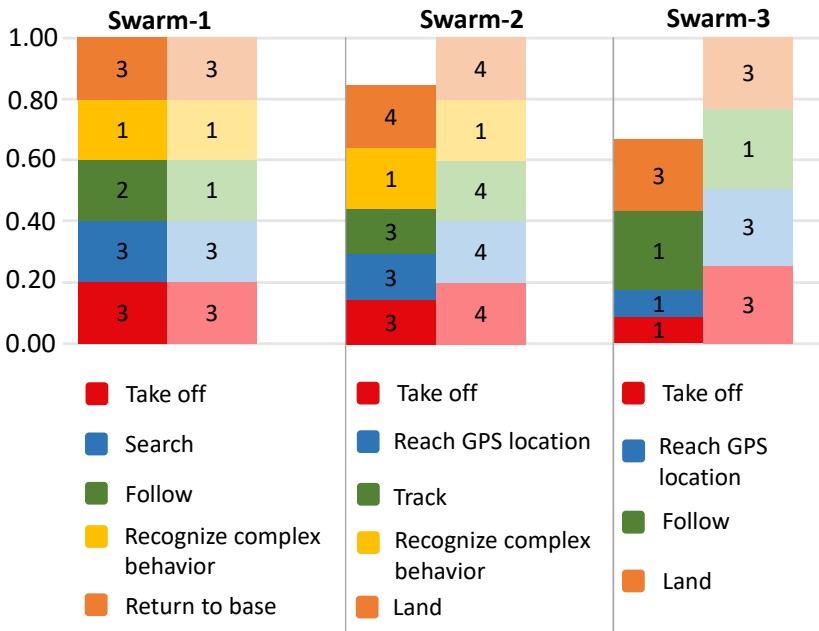

**Figure 6.** Multi-swarm performance. For each swarm, the right bar is the desired performance for each task, representing the minimum number of drones expected to complete each behavior, while the left bar is the actual performance, indicating how many drones successfully executed each task.

Additionally, the search path in swarm-1 is predefined, and that may reduce the search efficiency. Fortunately, the proposed architecture allows the modification and optimization

of their individual parts. Concretely, a more efficient method can be applied; for instance, if the drones are trained with a reinforcement learning algorithm to resourcefully distribute their search regions to minimize overlapping areas, as carried out in [45] over the same domain where these experimental results were conducted.

Moreover, the current mm-wave radar is employed using time-division multiplexing where only one Tx is transmitting at a time, resulting in a possible low signal-to-noise ratio (SNR) at the receiver end and causing poor detection accuracy if the object is too far away. Future radar architecture will use spatial multiplexing schemes such as binary-phase-modulation to perform the detection, where all the Txs are transmitting simultaneously to achieve a much higher SNR at the receiver end.

In addition to the mentioned improvements that come naturally from the test and results of the described experiment, further optimizations at each level of the proposed multi-layer policy will be considered in future applications, such as drone path planning, area exploration, collision avoidance, or task allocation, [45–48].

## 7. Conclusions

This paper showed an experimental test of a context-aware distributed team of SUAS coordinately working on a multi-step complex mission, capable of operating in real time, in an autonomous fashion, and under constrained communications. In this experiment, 10 drones divided into three teams performed the complex three-step task of (i) searching, detecting, and following a person until they entered a car, (ii) navigating to a specific GPS position and tracking a car until a person left the car, and (iii) navigating to a GPS position given by the previous team, and following a person at a constant distance for a period of time. The proposed framework relies on a three-layered approach: operational, tactical, and strategical, corresponding to single agent actions, collaboration between a group of agents, and joint cooperation of teams of multiple groups of agents, respectively. The complex mission was carried out based on the continuous loop perception–policy–decision architecture. The perception was based on the fusion of 4D RGB and 4D infrared cameras, together with 4D mmWave radar, and the communication among the agents in the teams was performed by an ad hoc network. The experimental validation showed that the complex task was propitiously achieved by the cooperation of the three teams. Although some agents in the teams may have not had the expected behavior due to possible packages loss, non-optimal illumination conditions for detection and tracking, and navigation issues due to uncertainties, the global behavior of the swarm managed to successfully complete the required mission.

**Author Contributions:** Conceptualization, J.M.-L.; Formal Analysis, J.M.-L.; Investigation, J.M.-L., Y.J., M.S. (Michael Shaham), Z.L., A.A.B., Y.W., W.Z., M.S. (Matthew Skopin), J.H.-J., Y.M., T.S., N.A. and A.F.; Methodology, J.M.-L.; Project Administration, J.H.; Resources, J.M.-L. and J.H.; Software, Y.J., M.S. (Michael Shaham), Z.L., A.A.B., Y.W., W.Z., Y.M., T.S., N.A. and A.F.; Supervision, J.M.-L. and J.H.-J.; Validation, J.M.-L., Y.J., M.S. (Michael Shaham), Z.L., A.A.B., Y.W., W.Z., M.S. (Matthew Skopin), J.H.-J., Y.M., T.S., N.A. and A.F.; Writing—original draft, J.M.-L. and J.H.-J.; Writing—review and editing, J.H.-J. All authors have read and agreed to the published version of the manuscript.

**Funding:** This work is funded by the U.S. Air Force Research Laboratory (AFRL), BAA Number: FA8750-18-S-7007.

**Data Availability Statement:** Not applicable.

**Conflicts of Interest:** The authors declare no conflict of interest.

## Abbreviations

The following abbreviations are used in this manuscript:

| | |
|---|---|
| CS | Compressive Sensing |
| GPS | Global Positioning System |
| MOT | Multiple Object Tracking |
| RGB | Red Green Blue |
| SNR | Signal-to-Noise Ratio |
| SUAS | Small Unmanned Aerial Systems |
| SUAV | Small Unmanned Aerial Vehicles |

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
