# Peer review of "Preliminary Experimental Results of Context-Aware Teams of Multiple Autonomous Agents Operating under Constrained Communications"

_robotics, doi:10.3390/robotics11050094_

Round 1
Reviewer 1 Report
This paper describes a framework of an autonomous multi-agent team consisting of small unmanned aerial systems. The framework contains three layers, covering the perception and decision in a top-down structure. Experimental demonstrations are provided.
This paper is poorly written. The presentations of the proposed framework are superfluous, where the descriptions stay with what it is: there is no analysis or deep discussion regarding why the proposed framework would work. Furthermore, given that multiagent systems are already well-studied, with abundant literature for the past ten years, leveraging unmanned aerial, ground, and even underwater vehicles, the authors didn't highlight what is new in the proposed approach. The literature review is too short to give the audience a clear idea of where the proposed work is compared to the vast amount of existing results. There are many poorly written sentences (for example, line 134-135), unnecessary math notation (U and J in lines 114-115), grammar mistakes (lines 124-125, "e.g., find survivors in ...") with low-quality figures (I can see the pixels in Figures 1 and 2). Besides, Figure 4 does not show any useful information except that multiple drones sit on the ground. The authors really should put more effort into writing their results clearly and cleanly.
For the reasons above, I do not recommend this paper to be published.
Reviewer 2 Report
The work is of scientific interest and is performed on a relevant topic,
associated with group control of drones. The work has applied significance.
However, there are a number of issues that could improve this work:
1. The methodology of algorithms for group control of drones is not sufficiently described.
2. Can one drone observe multiple objects at the same time?
3. How is a decision made to observe an object by a particular drone?
4. Are machine learning algorithms being used to improve the efficiency of group management?
Reviewer 3 Report
This paper proposes a framework for unmanned autonomous vehicles. However, the manuscript requires more explintation about some parts in proposed methodology. Here are the comments to atuhors,
1. At the operational layer, searching, detection, localization, 11 classifications, identification, tracking, and following of the person. There is no explanation of which models are proposed to perform detection? identification tracking and classification
2. It would be better to provide some numerical results about the detection tracking, classification, and real-time fps?
3. The authors use so many sensors, your method could be computationally very complex, however, there is no evidence for computational complexity.
4. There is a limitation of this work presented in the manuscript.
5. If your system is real-time, what are you sacrificing?
6. Figure 4 is not visible, is it possible to crop and provide a zoomed version of the same picture to be more clear.
7. No information about training, or testing is provided in the manuscript?
8. No loss functions and mathematical details of the proposed method.
Reviewer 4 Report
This paper presents a method for UAV coordination under imperfect communication conditions. The method is sound and proof-of-concept preliminary tests are also presented.
A general remark is that there is no mention in proposed method for optimisation of the decision process (see e.g. [1], [2]), something that would add value to the method and significantly enhance the scientific content of the paper. The authors are asked to comments on the proposed process optimisation and how they (intend to) handle it.
1. Yu, Wan-Yu, et al. "Auction-Based Consensus of Autonomous Vehicles for Multi-Target Dynamic Task Allocation and Path Planning in an Unknown Obstacle Environment." Applied Sciences 11.11 (2021): 5057.
2. Amanatiadis, Angelos A., et al. "A multi-objective exploration strategy for mobile robots under operational constraints." IEEE Access 1 (2013): 691-702.
Round 2
Reviewer 3 Report
The paper can be accepted in the current versions
Reviewer 4 Report
The authors have now substantially improved the manuscript.